# Improving YOLOv5 with Attention Mechanism for Detecting Boulders from Planetary Images

**Linlin Zhu [1]** **, Xun Geng [2,3], Zheng Li [1] and Chun Liu [1,3,*]**

1   School of Computer and Information Engineering, Henan University, Kaifeng 475001, China; linlinz@henu.edu.cn (L.Z.); lizheng@henu.edu.cn (Z.L.)
2   College of Geography and Environmental Science, Henan University, Kaifeng 475001, China; gengxun@henu.edu.cn
3   Henan Industrial Technology Academy of Spatio-Temporal Big Data, Henan University, Zhengzhou 450000, China
*   Correspondence: liuchun@henu.edu.cn

**Abstract:** It is of great significance to apply the object detection methods to automatically detect boulders from planetary images and analyze their distribution. This contributes to the selection of candidate landing sites and the understanding of the geological processes. This paper improves the state-of-the-art object detection method of YOLOv5 with attention mechanism and designs a pyramid based approach to detect boulders from planetary images. A new feature fusion layer has been designed to capture more shallow features of the small boulders. The attention modules implemented by combining the convolutional block attention module (CBAM) and efficient channel attention network (ECA-Net) are also added into YOLOv5 to highlight the information that contribute to boulder detection. Based on the Pascal Visual Object Classes 2007 (VOC2007) dataset which is widely used for object detection evaluations and the boulder dataset that we constructed from the images of Bennu asteroid, the evaluation results have shown that the improvements have increased the performance of YOLOv5 by 3.4% in precision. With the improved YOLOv5 detection method, the pyramid based approach extracts several layers of images with different resolutions from the large planetary images and detects boulders of different scales from different layers. We have also applied the proposed approach to detect the boulders on Bennu asteroid. The distribution of the boulders on Bennu asteroid has been analyzed and presented.

**Keywords:** planetary exploration; Bennu asteroid; boulder detection; YOLOv5; boulder distribution; attention mechanism; planetary image



## 1. Introduction

With the development of science and technology, more and more probes are launched to explore the planets. The probes take lots of images of the planets and provide valuable data for scientists. Because there are some typical objects such as the craters and boulders on the surfaces of planets, it is of great significance to detect these objects from planetary images and analyze their distribution. This will contribute to the selection of candidate landing sites and the understanding of the geological processes [1–5]. Among these objects, boulders are the frequent features on the surfaces of solid planets, especially for asteroids. For example, the OSIRIS-REx (Origins, Spectral Interpretation, Resource Identification, and Security-Regolith Explorer) mission confirmed that Bennu asteroid (101955) is a rubble pile dominated by boulders [6]. In light that the boulders are usually large in number and diverse in scale, it is beneficial to apply the object detection methods to detect the boulders in images of the planetary surface automatically.

In recent years, lots of novel object detection methods have been presented to detect objects from images with the tremendous success of deep learning. Generally, these methods can be divided into two categories: two-stage and single-stage. The two-stage

detection methods firstly extract the regions of interest (ROI) from the input images, and then carry out bounding box regression and classification within these ROIs. The single-stage detection methods view the object detection as a regression problem and do the localization and classification in the same stage. The two-stage detection methods have achieved good results in detection accuracy but have lower real-time performance. Meanwhile, the single-stage detection methods have slightly lower accuracy but higher detection speed. Thus, these single-stage detection methods are widely used in many studies. Among them, YOLOv5 [7] has achieved the state-of-the-art performance.

However, it is challenging to apply the object detection methods such as YOLOv5 directly to detect the boulders from the planetary images. First, the images of the planetary surface are often large in size, but the images fed into the YOLOv5 usually have a much smaller size. The large images of the planets need to be cut into smaller slices. This may lead to the large boulders being cut into several parts which are located in different slices. As a result, one boulder is wrongly recognized as several boulders. Second, the images returned by probes are often gray in color and the boulders in the images are blurred with the background due to the lack of contrast. Specifically, the boulders always pile up together. These also affect the performance of the object detection methods in detecting the boulders, particularly the small boulders from planetary images.

To address these challenges, this paper improves YOLOv5 detection methods with attention mechanism and proposes a pyramid based approach to detect multi-scale boulders from the planetary images. The attention mechanism is a way that enforces the learning process to focus on the important regions of the input objects such as the images by adjusting the weights to different regions. It is often used to obtain more critical information from the input objects. In our experiment, the input objects refer to the Bennu images or VOC2007 images to be detected. By introducing attention mechanism into YOLOv5 model, our aim is to enable the detection method to focus on the regions of interest in the images and enhance its performance in boulder detection. At the same time, the pyramid based approach aims to extract different layers of images with different resolutions from the large planetary image and detect the boulders of different scales from different layers. The main contributions of this paper are as follows:

- The YOLOv5 detection methods are improved by applying the attention mechanism. Besides three feature fusion layers which aggregate the feature maps of different levels to obtain more contextual information, an additional shallow layer is added to obtain more feature information of the small boulders. In addition, inspired by the idea of residual networks, new connections have been added to bring feature information from backbone network into these feature fusion layers to further reduce the feature information loss of the small boulders. Moreover, the attention modules implemented by combining the CBAM [8] and ECA-Net [9] attention mechanisms have been added into each feature fusion layer to highlight the information that contributes to the boulder detection. The evaluations have shown that these improvements have increased the performance of YOLOv5 by 3.4% in precision.

- A pyramid based approach is designed to detect multi-scale boulders. From the input large image, the proposed approach obtains several layers of images with lower resolution through downsampling. Then, these layers of images are cut into small slices which are fed into the improved YOLOv5 detection methods for boulder detection. Afterwards, the detection results of different layers are relocated to the original large image, and the non-maximum value suppression (NMS) [10] is used to filter the duplicate results. We have applied the proposed boulder detection approach to detect the boulders on the Bennu asteroid. The distribution of the boulders on Bennu asteroid has been also analyzed and presented.

The rest of the paper is organized as follows. Section 2 introduces the detection methods including the improved YOLOv5 and the pyramid based approach. Section 3 describes the datasets and the results, while Section 4 discusses the feasibility of the proposed approach. Section 5 is a summary of our work.

## 2. The Methods

In this section, we present the detection methods that we propose for detecting boulders from planetary images. We first give a brief introduction about the state of the art of the object detection methods based on deep learning, and then detail the YOLOv5 detection method and our improvements. Finally, the pyramid approach for detecting boulders from large planetary images is described.

### 2.1. The State of Art Object Detecting Methods Based on Deep Learning

As stated before, recent object detection methods can be divided into two categories: two-stage and single-stage. Typical two-stage object detection methods include R-CNN [11], SPP-Net [12], Fast R-CNN [13], Faster R-CNN [14], etc. R-CNN uses convolutional neural network for object detection. It uses the selective search algorithm to extract candidate frames, takes CNN to extract features from these candidate frames, and finally applies SVM classifier for classification. Although the detection accuracy is greatly improved compared to the traditional object detection methods, the method is more complicated and costs more time. In order to reduce the amount of calculation, Kaiming He et al. proposed SPP-Net, which introduced spatial pyramid pooling. It can extract features from arbitrary regions, making the network capable of detecting objects from the input images of various sizes. Later, the author of R-CNN designed Fast R-CNN, which adopted the idea of SPP-Net to segment the feature maps using rectangular boxes of specific size to obtain features in different regions, and used softmax [15] instead of SVM to classify the objects. In the same year, S. Ren et al. designed the Faster R-CNN network. Faster R-CNN proposed an anchor frame mechanism to generate the candidate frames and achieved faster candidate frame generation.

Typical single-stage object detection methods include the series of SSD detection methods and the series of YOLO detection methods. The series of SSD detection methods include SSD [16], DSSD [17], ESSD [18], MDSSD [19], etc., and the series of YOLO detection methods include YOLO [20], YOLOv2 [21], YOLOv3 [22], YOLOv4 [23], and YOLOv5 [7]. In 2015, Joseph R et al. proposed YOLO. The core idea is to segment each image into $7 \times 7$ sized squares. Each square is responsible for detecting and classifying the objects in that region. YOLO discards the step of generating suggestion boxes, reducing calculation and time-consumption. Since only the last layer of feature maps is used, the model has a poor detection performance on small objects. To improve the detection accuracy, Liu et al. in 2016 introduced an anchor frame mechanism in SSD while using VGG-16 [24] as a feature extraction network and designed an image pyramid structure so that the shallow feature maps and the high-level feature maps are used to detect small and large objects respectively. YOLOv2 and YOLOv3 also use an anchor frame mechanism, which improves the precision and detection speed at the same time. YOLOv2 uses Darknet-19, which borrows a priori frame from the RPN network [25], to maintain the detection speed while improving the accuracy of the model. YOLOv3 uses a 53-layer convolutional network for feature extraction to obtain three different sizes of feature maps. In 2017, Liu et al. proposed the DSSD object detection network. It uses ResNet101 [26] as a feature extraction network to extract deeper features, replaces the traditional bilinear interpolation upsampling with deconvolution, and introduces a residual module in the prediction phase. The detection accuracy is improved, but the speed shows a decline. In 2018, ESSD and MDSSD introduced high-level semantic information into shallow detail features and improved detection precision through multi-scale feature fusion. YOLOv4 uses CSPDarknet53 [22] as the backbone network and CIOU_LOSS [27] for prediction boxes screening, which improves the detection accuracy. YOLOv5 also uses CSPDarknet53, but the neck network has adopted the feature pyramid network(FPN) [16] and the pixel aggregation network (PAN) [28] structures. With a lightweight model size, it is comparable to YOLOv4 in terms of accuracy but superior to YOLOv4 in speed.

## 2.2. The YOLOv5 Method

The architecture of YOLOv5 is shown in Figure 1. It can be seen that YOLOv5 network consists of three parts: backbone part for feature extraction, neck part for feature fusion, and output part for object detection.

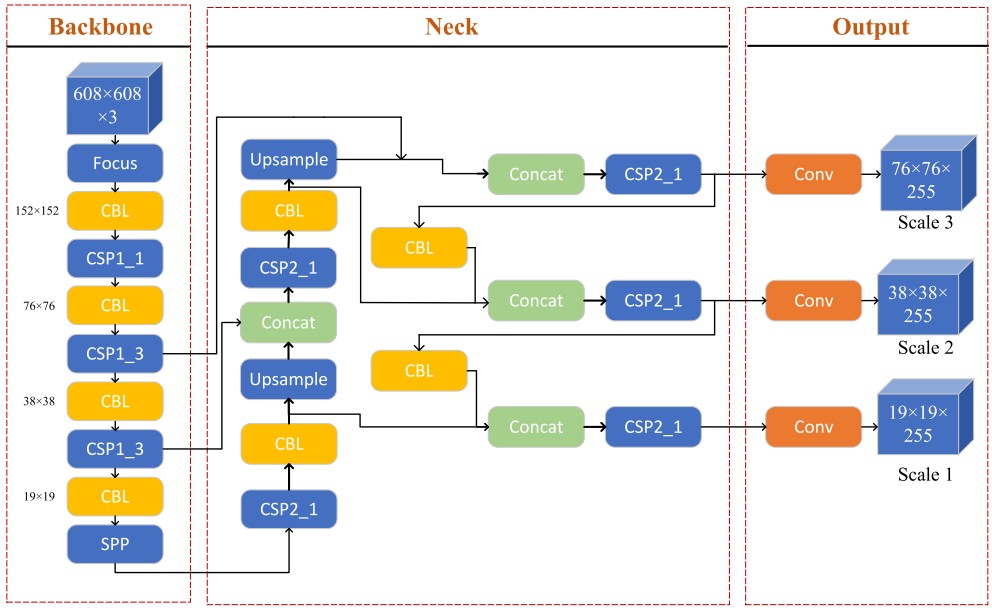

**Figure 1.** The architecture of the YOLOv5 method. The network consists of three main parts: backbone, neck, and output. Backbone part focuses on extracting feature information from input images, neck part fuses the extracted feature information and generates three scales of feature maps, and the output part detects the objects from these generated feature maps.

The backbone network is a convolutional neural network which extracts feature maps of different sizes from the input image by multiple convolution and pooling [29]. As shown in Figure 1, there are four layers of feature maps generated in the backbone network. The sizes of them are: 152 × 152 pixels, 76 × 76 pixels, 38 × 38 pixels, and 19 × 19 pixels. With these feature maps of different size, the neck network fuses the feature maps of different levels to obtain more contextual information and reduce the information loss. In the fusion process, the feature pyramid structures of FPN and PAN are used. The FPN structure conveys strong semantic features from the top feature maps into the lower feature maps. At the same time, the PAN structure conveys strong localization features from lower feature maps into higher feature maps. The two structures jointly strengthen the feature fusion capability of the neck network. Specially, it can be seen that there are three feature fusion layers which generate three scales of new feature maps with the sizes of 76 × 76 × 255, 38 × 38 × 255, and 19 × 19 × 255, where 255 indicates the number of channels. The smaller the size of the feature maps, the larger the area of the image that each grid unit in the feature map corresponds to. This indicates that it is suitable to detect large objects from the 19 × 19 × 255 feature maps, while the 76 × 76 × 255 feature maps are suitable for detecting small objects. From these new feature maps, the output network part performs object detection and classification.

In the architecture, the focus module slices the images and concatenates them, whose purpose is to better extract the features during downsampling. The CBL module consists of the modules of convolution, normalization, and Leaky_relu activation function [30]. Obviously, there are two kinds of cross-stage partial network (CSP) [31] in YOLOv5. One is used in backbone network and the other in neck. With cross-layer connectivity to connect the front and back layers of the network, CSP network aims to improve the inference speed while maintaining the precision by reducing model size. The two kinds of CSP networks have small difference in the structure. The CSP network in the backbone consists

of one or more residual units, while the CSP network in neck replaces the residual units with the CBL modules. Moreover, the SPP module means the spatial pyramid pooling module which executes the maximum pooling with different kernel size and fuses the features through concatenating them together. Pooling mimics the human visual system by performing dimensionality reduction (downsampling) operations to represent image features at a higher level of abstraction. It is mainly the compression of the input feature map. On the one hand, it makes the feature map smaller and simplifies the computational complexity of the network; on the other hand, it performs feature compression and extracts the main features. The Concat module means the tensor concatenation operation.

### 2.3. The Attention Mechanism

The attention mechanism is to obtain more critical information by focusing on the important regions of the input objects. In practice, there are different implementations for the attention mechanism in different applications. CBAM [8] is a lightweight widely used attention mechanism that combines spatial and channel attention. At the same time, ECA-Net [9] is an effective channel attention mechanism which can capture information about cross-channel interactions, i.e., the dependence between channels, and obtain a significant performance increase.

### 2.3.1. CBAM

The CBAM [8] is a combination of spatial and channel attention. The channel attention is to learn the weights of different channels and multiple the different channels with the weights to enhance the attention to the key channel domain. For the feature map of a layer $F \in R^{(C \times H \times W)}$ where C represents the number of channels, H and W represent the length and width of the feature map in pixels, channel attention module first calculates the weights of each channel $M_c \in R^{(c \times 1 \times 1)}$ according to the following formula.

$$M_c(F) = \sigma(W_1(W_2(F_{avg}^c)) + W_1(W_2(F_{max}^c))) \tag{1}$$

In above formula, $F_{avg}^c$ and $F_{max}^c$ represent the feature maps after average and maximize pooling, $W_1$ and $W_2$ represent the weights of two layers of a multilayer perception, and $\sigma$ is the sigmoid activation function. Then, the channel attention feature map is obtained by multiplying $M_c \in R^{(c \times 1 \times 1))}$ with the original feature map.

The channel attention feature map will be sent to the spatial attention module. The spatial attention focuses on the location information of the object on the images and selectively aggregates the spatial features of each space through the weighted sum of spatial features. Taking the channel attention feature map $F_c \in R^{(C \times H \times W)}$ as input, the maximize pooling and average pooling are performed successively as shown in Formula (2). Then, the spatial attention weight map $M_s \in R^{(1 \times h \times w)}$ is obtained through the convolution with the kernel of $7 \times 7$ as shown in Formula (3).

$$F_s = \frac{1}{c} \sum_{i \in c} F_c(i) + \max_{i \in c} F_c(i) \tag{2}$$

$$M_s = \sigma(f^{(7 \times 7)}(F_s)) \tag{3}$$

After that, the spatial attention weight map and the input channel attention feature map are multiplied to obtain the final attention feature map.

### 2.3.2. ECA-Net

The channel attention in CBAM compresses the spatial dimensionality of the feature map through global average pooling and maximum pooling to capture nonlinear cross-channel interactions, which involves dimensionality reduction to control the complexity of the model. However, dimensionality reduction has side effects on capturing the dependence between all channels. The ECA-Net mechanism [9] avoids dimensionality reduction by adding a small number of parameters but effectively captures information

about cross-channel interactions and obtains a significant performance increase. After channel-level global average pooling without dimensionality reduction, ECA-Net captures local cross-channel interaction information by considering each channel and its *K* neighbors. The convolutional kernel size *K* represents the coverage of local cross-channel interactions, i.e., how many neighbors of that channel are involved in the attention calculation. To avoid manual tuning of *K*, a method is used to adaptively determine *K* as shown in Formula (4).

$$K = \varphi(C) = \left| \frac{\log_2(c)}{r} + \frac{b}{r} \right|_{odd} \tag{4}$$

Here, *C* is the channel dimension, $|t|_{odd}$ denotes the nearest odd number of *t*, *r* is set to 2 and *b* to 1.

### 2.4. Improved YOLOv5 with Attention Mechanism

To detect multi-scale boulders from the planetary images, we improve the YOLOv5 detection method with attention mechanism. As shown in Figure 2, there are three improvements to the original YOLOv5 architecture: (1) a new feature fusion layer marked with the green background is added to capture more shallow feature information of small boulders; (2) the features from the backbone network are brought into the feature fusion layers (represented by the red lines) to reduce feature information loss of small boulders; (3) and the attention modules shown with purple color are added into the fusion layers to highlight the information that contributes to the boulder detection.

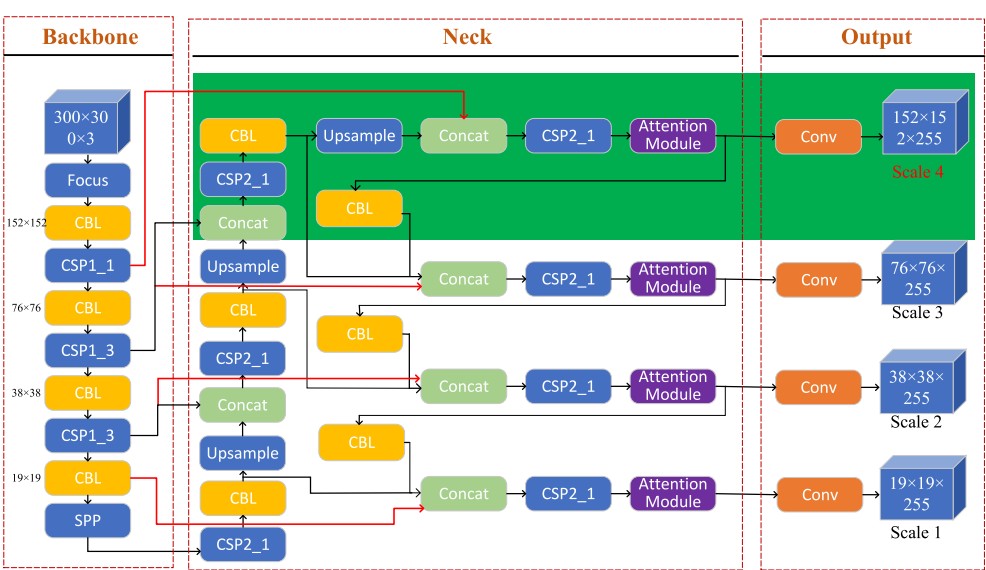

**Figure 2.** The architecture of the improved YOLOv5 method. Compared to original YOLOv5 method, there are three improvements in the architecture. First, a new fusion layer is added which generates a large scale of feature map with the size of 152 × 152 × 255. Second, new connections represented by red lines have been added to bring feature information from backbone into feature fusion layers. Third, the attention modules are added into feature fusion layers.

First, to improve the performance of YOLOv5 on detecting the small boulders, a new fusion layer is added which generates a larger feature map with the size of 152 × 152 × 255. Compared to the original YOLOv5 model in Figure 1, there are four fusion layers in the improved YOLOv5. The structure of the new fusion layer can be seen in Figure 2. On the basis of the original network, the fused feature maps are further unsampled and concatenated with the feature map of 152 × 152 pixels from the backbone network to generate a new layer of fused feature maps. In this process, the CSP modules and the CBL modules are also used.

Second, four connections represented by the red lines are added to bring the feature information from the backbone network (152 × 152 pixels, 76 × 76 pixels, 38 × 38 pixels,

19 × 19 pixels) into the feature fusion layers in the neck network. Based on the idea of residual networks, these connections can enhance the back propagation of gradients, avoid gradient fading, and reduce the loss of the feature information of small objects.

Third, in order to amplify the information of the boulders in the images, the attention modules are added into feature fusion layers. As shown in Figure 3, the attention modules are implemented as a combination of the CBAM and the ECA-Net. That is, the channel attention is selected from ECA-Net [9] and the spatial attention is from CBAM [8]. The ECA module first learns directly on the features after global average pooling (GAP) by 1D convolution and multiplies the updated weights with the input feature map to generate a new feature map. The feature map generated by the ECA module is used as the input of the spatial attention module of CBAM. Through spatial attention module, the spatial attention feature map will be generated and summed with the original feature map to simulate the residual block structure. In the end, the final feature map is output after applying the Relu activation function to the summed feature map.

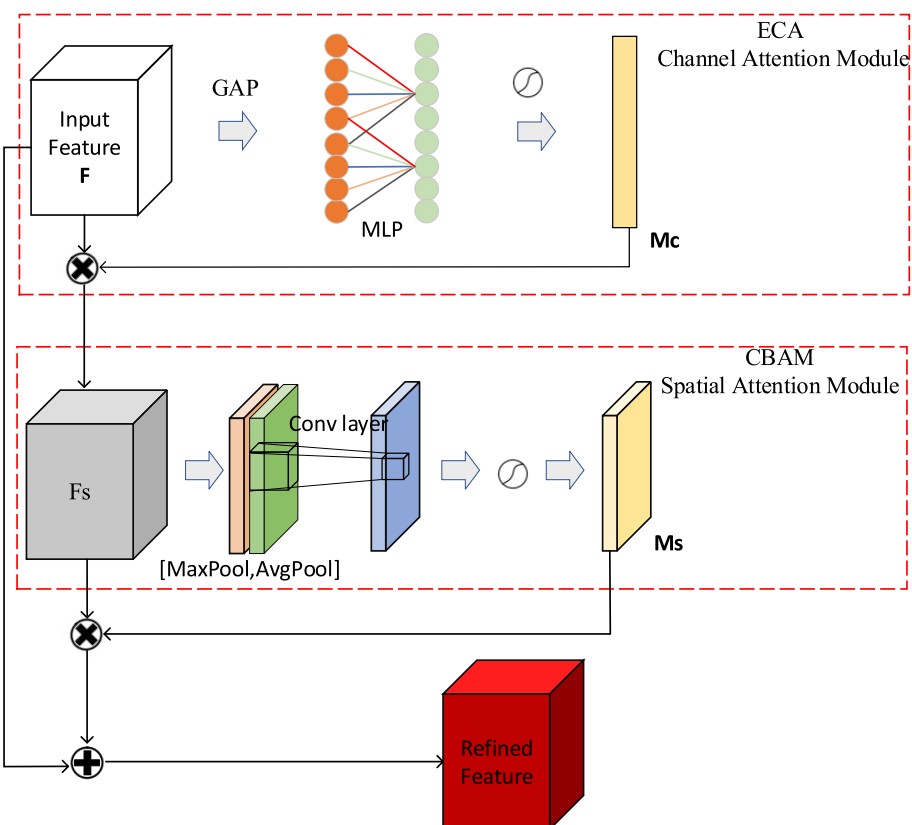

**Figure 3.** The structure of the attention modules. The attention modules are implemented by combining the ECA and the CBAM attention mechanisms. The above part is the ECA attention module which implements the channel attention, and the blow part is the CBAM spatial attention module.

### 2.5. The Pyramid Based Approach Using Improved YOLOv5 for Detecting Boulders

In light of the large size of the images from the planets and the diverse scales of the boulders in the images, an image pyramid based approach is designed in this paper for detecting boulders based on the improved YOLOv5 detection method. The basic idea is to construct an image pyramid from the large image and then detect the objects from each layer with different spatial resolutions to improve the detection precision. The framework of the proposed pyramid based approach is shown in Figure 4, which consists of four steps: Pyramid Construction, Hierarchical Slice, Boulder Detection, and Results Relocation.

Taking a large image as input, the first step is to construct an image pyramid by extracting several layers of images with different resolutions from the input large image through downsampling. Because the images fed into the YOLOV5 usually have a much

smaller size such as 300 × 300 pixels, we need to cut the input large images into a set of small images for YOLOv5. However, the cutting process may cut the large boulders into several parts which are located in different small images and leads to one boulder being wrongly recognized as several boulders. The purpose of constructing image pyramid is to avoid that case. For the large boulders in a lower layer of the image pyramid, they will become smaller in the higher layer of the image pyramid. Thus, the large boulders can be accurately recognized from the image of higher layer. Furthermore, through relocating the detection results of each layer into the original input image, the wrongly detected results generated in the lower layers will be eliminated with the NMS algorithm. The number of layers extracted is determined according to the size of the original image and the size of the input of YOLOv5 method. The size of the top layer should be less than four times the input size of YOLOv5 method. Otherwise, once the size of one generated layer is larger than four times the input size of YOLOv5 method (both the width and height should be more than twice the width and height of the input image of YOLOv5 respectively), a new higher layer will be generated.

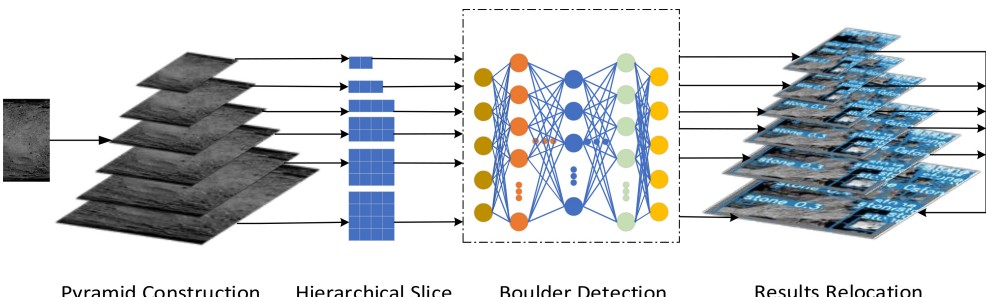

Pyramid Construction    Hierarchical Slice    Boulder Detection    Results Relocation

**Figure 4.** The framework of the pyramid based approach.

The second step is to cut the images of each layer of the pyramid into small slices which can be fed into the improved YOLOv5 method. The images of each layer are cut separately according to the size of the input of YOLOv5 method. The third step is the boulder detection process in which the boulders are detected by applying the improved YOLOv5 method. We feed each slice into YOLOv5 and use it to predict the position of the boulders on the slices. The fourth step is to relocate the detection results into the original input image and eliminate the duplicate ones. Actually, there are two relocation steps. One is to relocate the detection results from each slice to their corresponding layer. Another is to further relocate the detection results of each layer to the original input image. Once all the detection results are mapped to the original input image, the NMS algorithm is applied to eliminate the duplicate detection results.

## 3. The Datasets and Results

To evaluate the improved YOLOv5 method, two datasets have been used. One is the VOC2007 [32] which is widely used for the evaluations of detection methods, and the other is the dataset constructed from the images of Bennu asteroid provided by OSIRIS-REx. We have also applied the proposed pyramid based approach to detect the boulders on the surface of Bennu asteroid and analyze their distribution. In this section, we describe the datasets, the implementations and settings, and the results of the evaluation. The distribution of the boulders on Bennu asteroid is also presented.

### 3.1. Datasets

To test the performance of the improved YOLOv5 on the detection of boulders, we have constructed a boulder dataset from the images of the Bennu asteroid. The Bennu asteroid is currently more than 200 million miles away from Earth. The diameter of Bennu's equator is about 492 m. The Lincoln Near-Earth Asteroid Research Team discovered the Bennu asteroid on 11 September 1999 [33]. The global map of the surface of the Bennu asteroid was created by stitching together images collected by NASA's OSIRIS-REx space-

craft between 7 March and 19 April 2019. A total of 2,155 PolyCam images were stitched together and corrected to make a mosaic. At 2 inches (5 cm) per pixel, this is the highest resolution available globally for mapping planetary bodies. The spacecraft collected these images at a distance of 1.9 to 3.1 miles (3.1 to 5 km) from the surface of the asteroid [34].

The region of 121°E–178°E, 90°S–88°N in the global map of the Bennu asteroid was selected as the dataset for model training, and we cut the selected part into small slices of 300 × 300 pixels in 8-bit JPG format. As a result, there are 884 slices in total. For these small slices, some of them do not have boulders. Therefore, we have removed these slices in which there are none boulders manually. These images without boulders are not used to train the model but are used for the testing sessions. Finally, there are 729 slices that remain. For these slices, the boulders are labeled manually by using the tool of LabelImg. During the marking process, we mark as many boulders that can be recognized by human eyes as possible. In practice, the dataset for YOLOv5 is enhanced with Mosaic technique. That is, the dataset will be enriched by the operations such as random scaling and stitching. To train YOLOv5 model for detecting boulders, the enriched dataset is divided into training set and test set according to the ratio of 9:1.

Besides the boulder dataset, the dataset of VOC2007 is also used to further test whether the improved YOLOv5 has better performance on the commonly used dataset. VOC2007 contains 9963 labeled images, with 24,640 objects labeled in 20 categories, such as people, cars, dogs, chairs, etc. It is a benchmark for the evaluations of the image classification and recognition methods. It has been widely used for the evaluations of the detection methods such as Faster R-CNN [14], YOLO [20], and YOLOv2 [21]. In our experiments, the VOC2007 dataset is divided into training and test sets in the same way as the Bennu asteroid dataset.

### 3.2. The Implementations and Settings

The implementation of YOLOv5 [7] comes from GitHub which is written in Python language. The training was run under Linux system, CUDA version 10.1, Pytorch version 1.6.0, Python version 3.8, and NVIDIA Tesla T4. We have trained the detection methods for 300 epochs by using the Adam optimizer [35] with a learning rate of 0.01 and a batch size of 16. Moreover, the confidence threshold is set to 0.25, which means that objects with a similarity of 0.25 or above will be marked.

### 3.3. The Comparison with Related Object Detection Methods

To validate the performance of the proposed improved YOLOv5, we have compared it with the related detection methods including SSD [16], ESSD [18], MDSSD [19], YOLOv3 [22], YOLOv4 [23], and YOLOv5 [7]. The precision and the FPS (frames per second) are used as measurement indicators. Precision is calculated as the proportion of the number of positive samples correctly predicted to the number of samples predicted as positive samples. It is defined as follows:

$$Precision = \frac{TP}{TP + FN} \tag{5}$$

The TP and FN represent the number of predictions of true positive samples and the number of predictions of false negative samples which are negative samples but predicted as positive samples. When the boulder prediction category is correct and the intersection over union (IoU) [36] measuring the ratio of the intersection between the prediction box and the ground truth is larger than a threshold (0.6 in our experiments), the detection is considered to be correct. FPS indicates the number of images that can be processed by the object detection methods in each second, which examines the real-time performance of the detecting methods. The higher the FPS value, the faster the detection speed. The comparison results are shown in Table 1.

From the results in Table 1, it can be seen that the improved YOLOv5 outperforms the other detection methods. For example, it outperforms the YOLOv5 by 3.4% on the boulder dataset and 1.5% on the VOC2007 dataset. These results show that the improved YOLOv5 detection method achieves the best result in terms of precision. However, in terms of FPS, the detection speed has dropped compared to the original YOLOV5 model.

**Table 1.** The comparison results with related methods.

| Dataset | Method | Backbone | Precision | FPS |
|---------|--------|----------|-----------|-----|
| Bennu | SSD | VGG-16 | 44.1% | 11 |
| | ESSD | VGG-16 | 43.0% | 11 |
| | MDSSD | VGG-16 | 53.0% | 4 |
| | YOLOv3 | Darknet-53 | 59.4% | 33 |
| | YOLOv4 | CSPDarknet53 | 72.0% | 32 |
| | YOLOv5 | CSPDarknet53 | 73.2% | 30 |
| | Ours | CSPDarknet53 | 76.6% | 25 |
| VOC2007 | SSD | VGG-16 | 77.5% | 46 |
| | ESSD | VGG-16 | 79.4% | 25 |
| | MDSSD | VGG-16 | 78.6% | 28 |
| | YOLOv3 | Darknet-53 | 74.5% | 36 |
| | YOLOv4 | CSPDarknet53 | 78.1% | 35 |
| | YOLOv5 | CSPDarknet53 | 82.7% | 36 |
| | Ours | CSPDarknet53 | 84.2% | 28 |

*3.4. The Analysis of the Boulders on Bennu Asteroid*

To validate the proposed approach, we have applied it to detect the boulders on global map of the Bennu asteroid and analyze their distribution. In this section, we present the results of the boulder detection and analysis, including the total number of the boulders in different regions of the Bennu asteroid, and the number of boulders of different scales.

As the size of the entire Bennu asteroid image is 31,417 × 15,709 pixels, a pyramid consisting of six layers is constructed. Accordingly, as shown in Table 2, there are 4802 slices of 320 × 320 pixels for the first layer, 1176 slices of 320 × 327 pixels for the second layer, 288 slices of 327 × 327 pixels for the third layer, 72 slices of 327 × 327 pixels for the fourth layer, 18 slices of 327 × 327 pixels for the fifth layer, and 3 slices of 327 × 491 pixels for the sixth layer. Moreover, there are no overlapping areas between the individual slices of each layer. Table 2 also shows the number of boulders detected from each layer and the final detection result. We can see that the final detection result of boulders is less than the result of first layer. This is because when generating the slices of the first layer, the large boulders may be cut into several parts which are wrongly recognized as boulders as well. These boulders wrongly recognized will be filtered in the proposed pyramid based approach.

**Table 2.** The number of boulders detected from the images of Bennu asteroid.

| Layer | Layer Size | Slice Size | Slices Num | Boulders Num Detected | Result |
|-------|-----------|-----------|-----------|----------------------|--------|
| First layer | 31,417 × 15,709 | 320 × 320 | 4802 | 276,431 | |
| Second layer | 15,709 × 7855 | 320 × 327 | 1176 | 99,812 | |
| Third layer | 7855 × 3928 | 327 × 327 | 288 | 19,768 | 257,238 |
| Fourth layer | 3928 × 1964 | 327 × 327 | 72 | 3562 | |
| Fifth layer | 1964 × 982 | 327 × 327 | 18 | 510 | |
| Sixth layer | 982 × 491 | 327 × 491 | 3 | 21 | |

### 3.4.1. Statistics of the Number of Boulders

From Table 2, it can be seen that 257,238 boulders are identified from the image of Bennu asteroid. To further understand these boulders, we have estimated the size of each boulder by using the diameter of the boulders generated by the improved YOLOv5 detection method. We have sorted the 257,238 detected boulders by diameter size. We divided the boulders into four scales according to diameter and counted the number of boulders in each scale, as shown in Figure 5. The horizontal coordinate indicates the diameter of the boulder, and we use "D" to represent the diameter in meters (m).

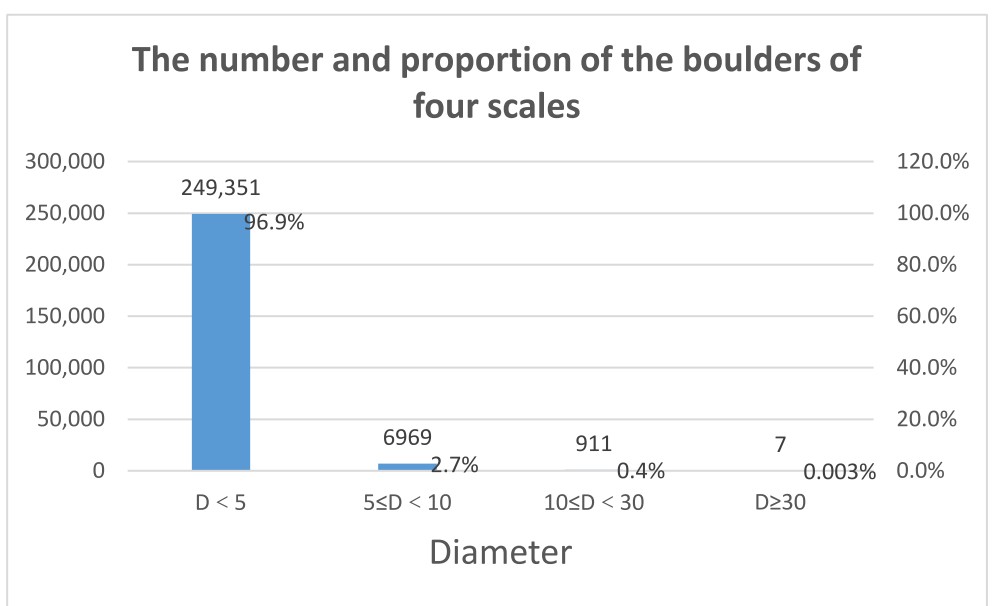

**Figure 5.** The number and proportion of the boulders of different scales.

The results in Figure 5 show that most of the detected boulders are less than 5 m in diameter. More accurately, there are 249,351 boulders whose diameters are less than 5 m, which accounts for 96.9% of the total number of detected boulders. The boulders whose diameters are more than 5 m accounts for 3.1% of the total boulders. The number of the large boulders whose diameters are more than 30 m is only 7.

### 3.4.2. Boulder Distribution Statistics

In order to analyze the distribution of the boulders on the Bennu asteroid, we have statistically analyzed the number of the detected boulders in the regions of 40°N–88°N, 0°–40°N, 0°–45°S, 45°S–90°S. Figure 6 shows the number of boulders in these regions. It can be seen that there are 34,472 boulders detected in the 40°N–88°N region, 71,299 boulders in the 0–40°N region, 87,065 boulders in the 0°–45°S region, and 64,402 boulders in the 45°S–90°S region. By comparison, the region of 0°–45°S has most boulders and accounts for about 34% of the total number of boulders. The region of 40°N–88°N has the fewest boulders and accounts for 13%. In addition, the northern hemisphere accounts for about 41% of the total number of boulders, and at the same time the southern hemisphere accounts for about 59% of the total number of boulders. This means that compared to northern hemisphere, there are more boulders in the southern hemisphere of Bennu asteroid. Furthermore, the 45°S–40°N region accounts for 62% of the total amount of boulders, indicating that most boulders are distributed near the equator. These results show the boulder distribution on the surface of Bennu asteroid. Furthermore, the boulders detected are marked by the yellow boxes.

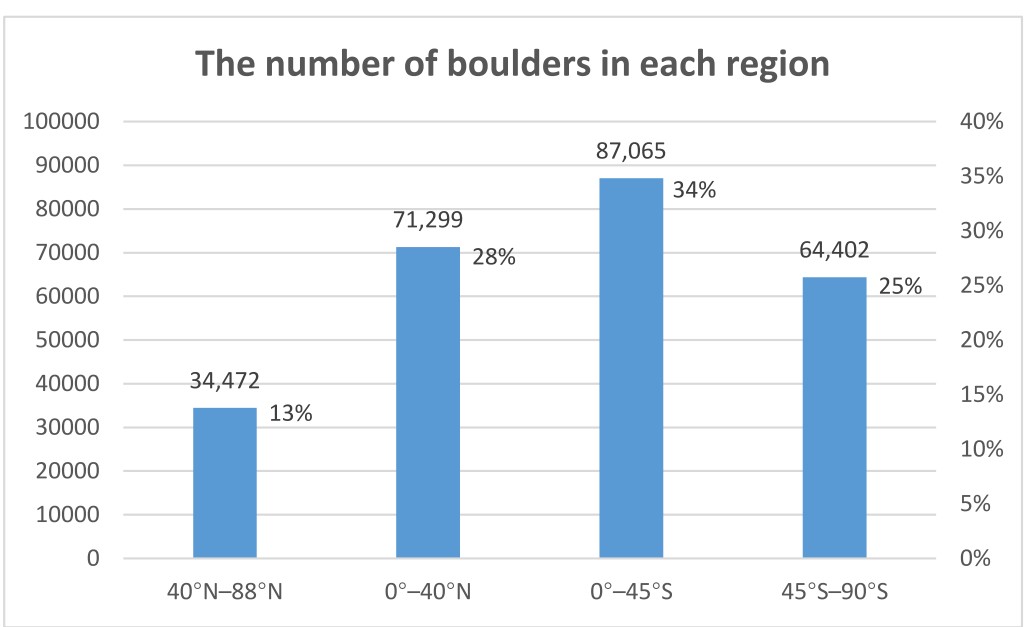

**Figure 6.** The number and proportion of boulders in each region.

Figure 7 shows the distribution of the boulders whose diameters are less than 5 m. The upper subfigure shows the number of these boulders in each region, and the subfigure below shows the distribution of these boulders across the surface of the Bennu asteroid. It can be seen that there are 155,226 boulders with the diameters less than 5 m detected in the region of 40°N–45°S, accounting for about 62% of the total number. This indicates that most of the boulders whose diameters are less than 5 m are near the equator. In addition, the region of 40°N–88°N has the lowest number of boulders with the diameters are less than 5 m, accounting for only 13% of the total number. These are consistent with that shown in the subfigure below Figure 7. From the subfigure below Figure 7, we can see that boulders less than 5 m in diameter are more densely distributed, especially in the southern hemisphere. Among these detected boulders whose diameters are less than 5 m, there is a proportion of buoulders which are less than 2.5 m in diameter (pixel values less than 20 pixels). We infer that there may be false detections in boulders smaller than 5 m. This is partly due to the fact that our detection precision was only 76.6%. This may be also partly caused by that we mark as many boulders visible to the human eye as possible and some of them may be small in diameter. In fact, it is difficult to determine which stones are not the boulders while zooming into the images to mark the boulders.

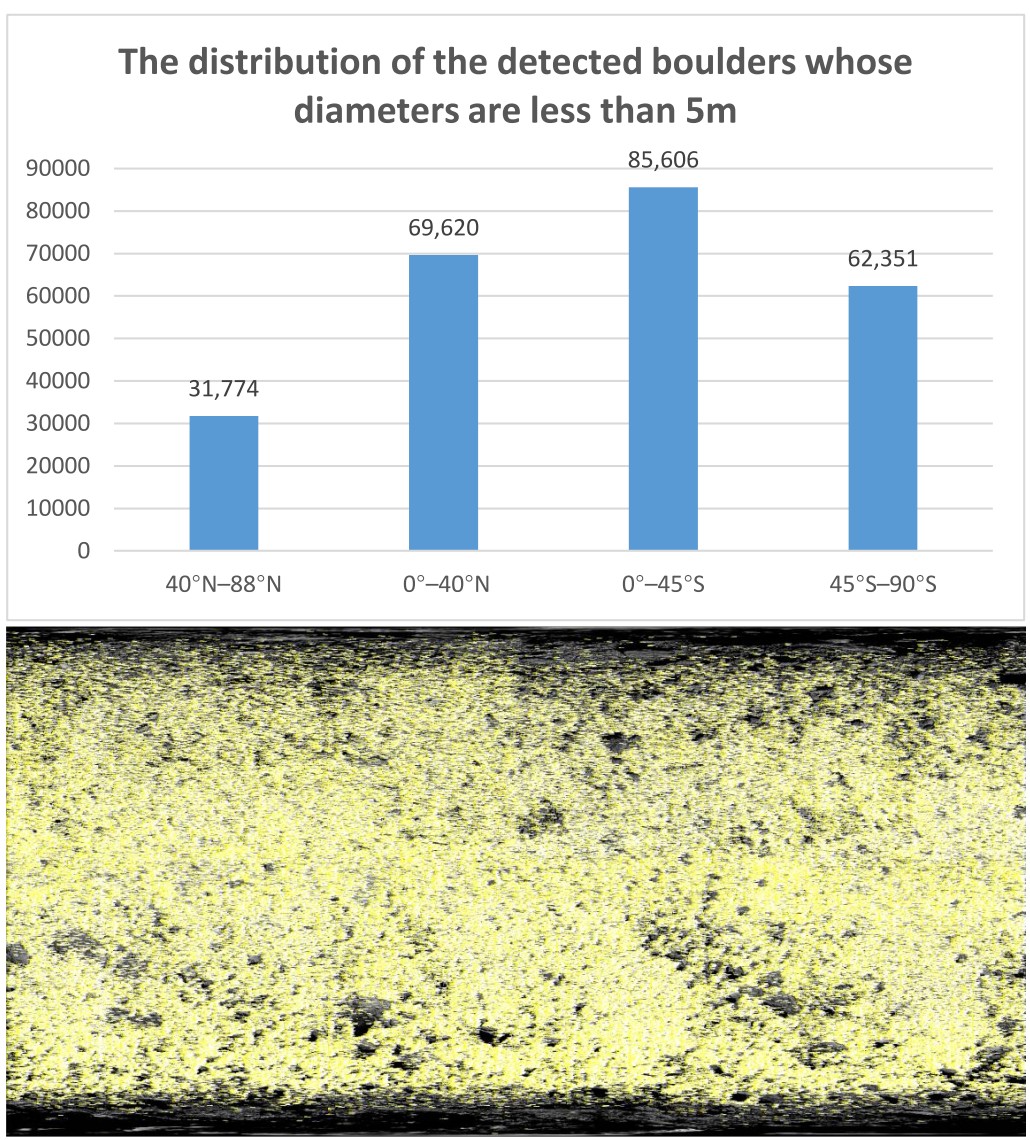

**Figure 7.** The distribution of the detected boulders whose diameters are less than 5 m in each region.

Figure 8 shows the distribution of boulders whose diameters are between 5 m and 10 m in each region on the surface of the Bennu asteroid. For the numbers shown in the subfigure in the up of Figure 8, it can be seen that there are 4082 detected boulders whose diameters are between 5 m and 10 m at the poles, i.e., the regions of 40°N–88°N and 45°S–90°S, accounting for about 59% of the total number. This shows that more boulders of this kind are located at the poles. It can be also seen from that shown in the subfigure below Figure 8.

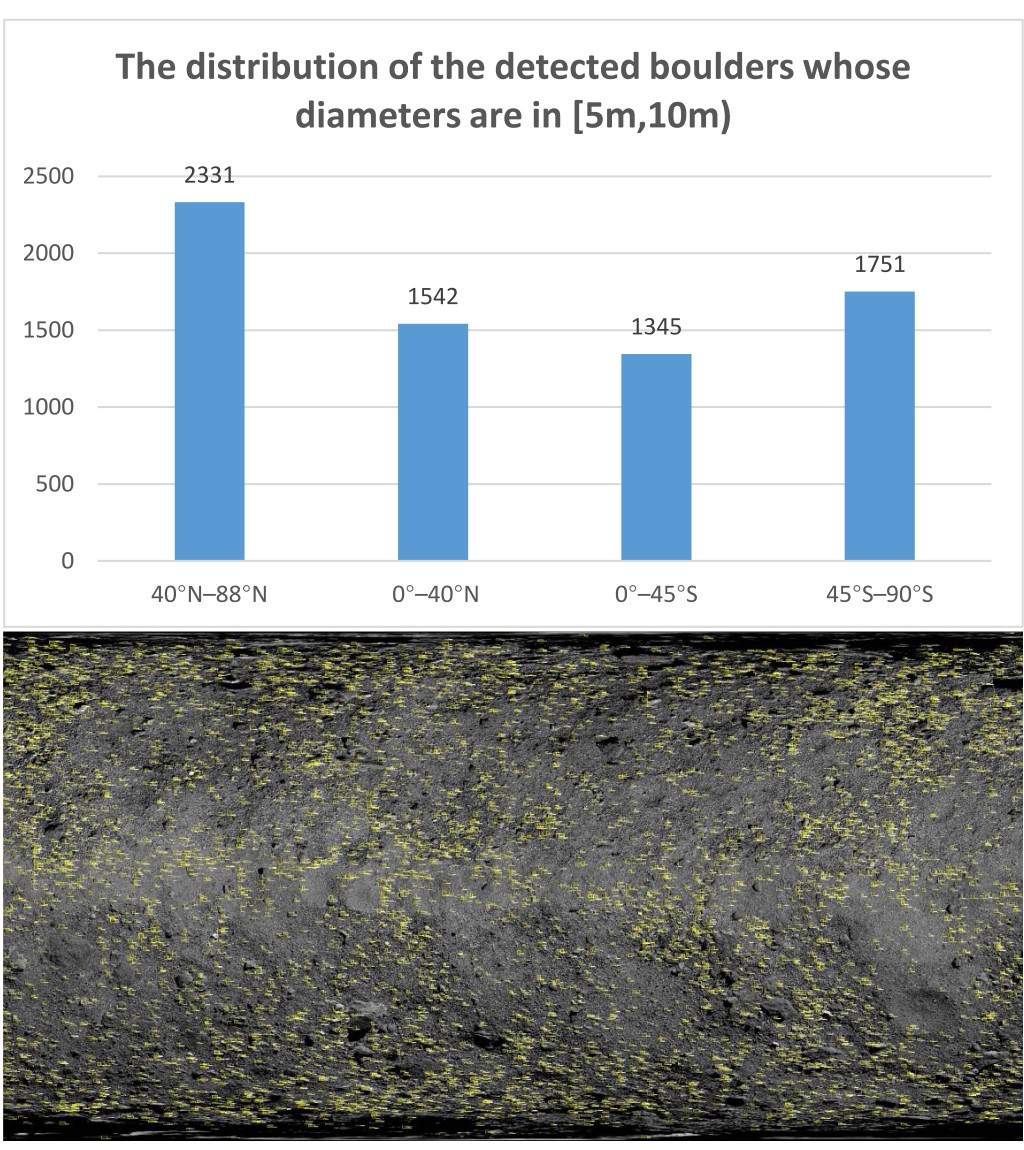

**Figure 8.** The distribution of the detected boulders whose diameters are between 5 m and 10 m in each region.

Figure 9 shows the distribution of the boulders whose diameters are between 10 m and 30 m in each region on the surface of the Bennu asteroid. There are 661 boulders with the diameters between 10 m and 30 m detected in the regions of 40°N–88°N and 45°S–90°S, accounting for about 73% of the total number. From the subfigure below Figure 9, it can be seen that the boulders are also slightly more numerous at the poles compared to these near the equator.

Figure 10 shows the distribution of the boulders whose diameters are more than 30 m in each region on the surface of the Bennu asteroid. It can seen that there are a total of seven boulders whose diameters are more than 30 m detected, including four in the southern hemisphere and three in the northern hemisphere.

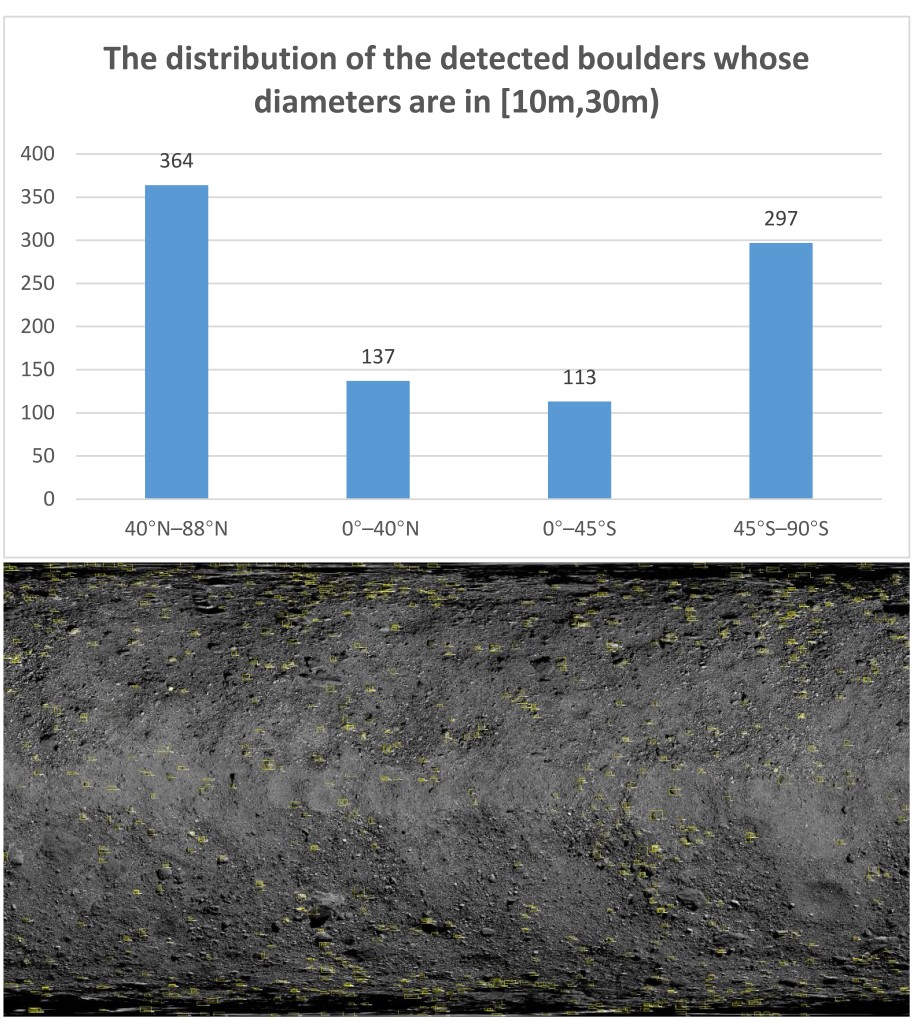

**Figure 9.** The distribution of the detected boulders whose diameters are between 10 m and 30 m in each region.

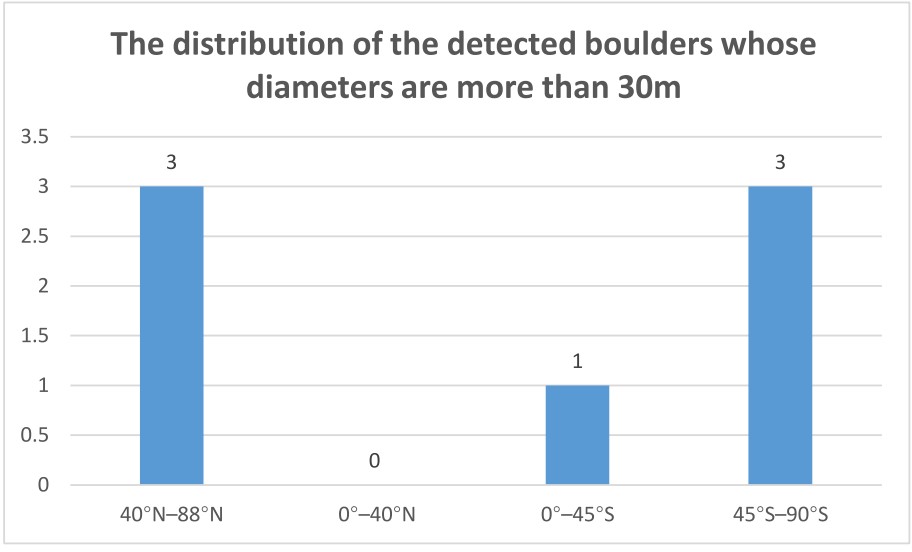

**Figure 10.** *Cont.*

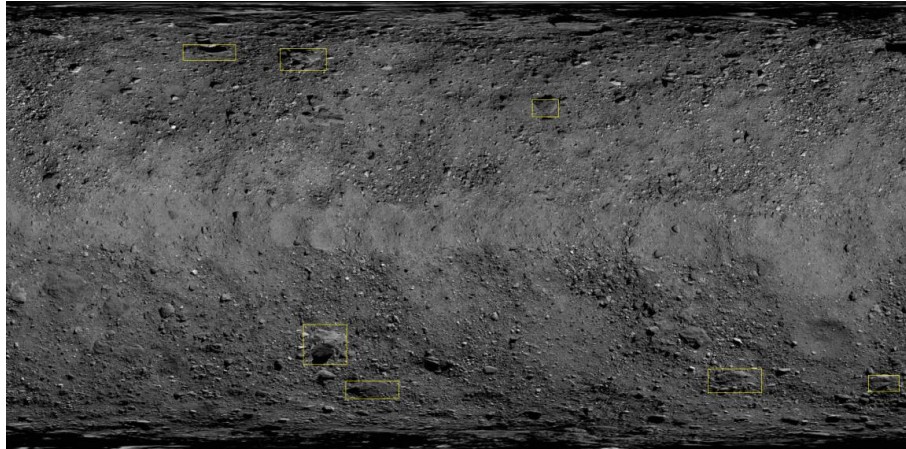

**Figure 10.** The distribution of the detected boulders whose diameters are more than 30 m in each region.

## 4. Discussion

### 4.1. Applications and Limitations

The method proposed in this paper focuses on the automatic detection of boulders in planetary images, which is promising for the analysis of the distribution of the boulders on the surface of planets. We evaluate the proposed method on the Bennu asteroid dataset labeled by us and on a typical object detection dataset—VOC2007. The results show that our improved YOLOv5 method not only improves the precision of the boulder detection from planetary images but also has better performance on the common object detection dataset. This demonstrates the superiority and applicability of our proposed method.

However, it should be noted that the proposed method still has some limitations. For example, although our method has improved precision compared to other methods, the precision only reaches 76.6%. This means more improvements are required in the future. Moreover, while improving the detection precision, it can be seen that the detection speed has dropped compared to other methods. The detection speed of our method is only close to 25 FPS on the Bennu asteroid dataset and 28 FPS on the VOC2007 dataset. This has not reached the standard of real-time detection.

### 4.2. Hyperparameter Exploration

There are several training parameters which can affect the performance of the deep learning model trained. These parameters include the size of input images, the number of epochs, the batch size representing the number of images input to the model each time, the learning rate, and the optimizer used. In our experiments, we have trained the detection methods for 300 epochs by using the Adam optimizer [35] with a learning rate of 0.01 and a batch size of 16. Whether these settings are optimal need to be verified. Therefore, we have done several experiments by adjusting these parameters and observed the performance changes of YOLOv5 based on the Bennu asteroid dataset. The experimental results are shown in Table 3. The "exp1-exp8" in Table 3 represents the id of the experiments, the "Img size" represents the size of the input image, the "Epochs" is the number of training iterations, the "Batch size" is the number of samples input to the network each time, and the "LR" is the learning rate.

In the experiments, we found that the loss tended to be stable when the epoch was close to 300, so the epoch is set to 300. From the Table 3, we find that exp8 has the highest precision in which the batch size is set to 16, the learning rate to 0.01, and the Adam optimizer [35] which is more suitable for the small dataset that is used in the training. This verifies our settings to these hyperparameters.

**Table 3.** Parameter adjustment and results.

| Number | Img Size | Epochs | Batch Size | LR | Optimizer | Precision |
|--------|----------|--------|------------|------|-----------|-----------|
| exp1 | 640 | 300 | 16 | 0.01 | Adam | 72.0% |
| exp2 | 320 | 300 | 16 | 0.01 | SGD | 71.5% |
| exp3 | 320 | 300 | 16 | 0.001 | Adam | 71.7% |
| exp4 | 320 | 300 | 16 | 0.1 | Adam | 72.4% |
| exp5 | 320 | 300 | 8 | 0.01 | Adam | 72.1% |
| exp6 | 320 | 300 | 32 | 0.01 | Adam | 71.2% |
| exp7 | 320 | 300 | 64 | 0.01 | Adam | 70.1% |
| exp8 | 320 | 300 | 16 | 0.01 | Adam | 73.2% |

*4.3. The Ablation Study*

Ablation study is a labour-saving way to study causality. In the case of complex deep neural networks, the performance of the network is usually studied by removing parts of the network in order to better understand the behavior of the network. We have put forward three improvements to YOLOv5 method. Whether all these improvements work well and whether there is interaction between them also need be investigated. We have run improved YOLOv5 with different improved settings on the Bennu asteroid dataset and recorded their performance in Table 4.

**Table 4.** The results of ablation study. In the first column, the "YOLOv5" detection method in the first row refers to the original YOLOv5 model; the "improved YOLOv5" in the second row refers to the model that adds the new feature fusion layer to YOLOv5; the third row refers to the model that further adds new connections into YOLOv5 to bring shallow features from backbone network for feature fusion; the fourth row refers to the model that applies ECA attention mechanism to model of the third row; the fifth row refers to the model that applies the CBAM attention mechanism to the model of the third row; and the sixth row refers to our proposed model that applies the combined ECA and CBAM attention mechanism to the model of third row.

| Method | Scale4 Layer | New Connections | Attention Mechanism | | | Precision |
|--------|--------------|-----------------|------|------|----------|-----------|
| | | | ECA | CBAM | ECA+CBAM | |
| YOLOv5 | × | × | × | × | × | 73.2% |
| Improved YOLOv5 | ✓ | × | × | × | × | 75.1% |
| Improved YOLOv5 | ✓ | ✓ | × | × | × | 75.5% |
| Improved YOLOv5 | ✓ | ✓ | ✓ | × | × | 76.1% |
| Improved YOLOv5 | ✓ | ✓ | × | ✓ | × | 73.7% |
| Ours | ✓ | ✓ | × | × | ✓ | 76.6% |

As shown in Table 4, the precision for YOLOv5 is 73.2% (first row). Then we evaluated the effectiveness of the new feature fusion layer of Scale4 and obtained the precision of 75.1% (second row). This shows that the new feature fusion layer works and contributes to boulder detection. Next, we tested the new connections to bring shallow features from backbone network for feature fusion. The precision reached 75.5% (third row). This also means that the added connecting lines have a positive effect on the detection of boulders. Finally, we tested the effectiveness of the attention modules added. When we applied the ECA-Net attention modules to YOLOv5, the precision value reached 76.1% (forth row). When we applied the CBAM attention modules to YOLOv5, the precision value was 73.7% (fifth row). This shows that there are interactions between the new feature fusion layer, the new connections and the CBAM attention mechanism, and they may conflict with each other. However, when we combined ECA-Net with CBAM to implement the attention

modules, the precision reached 76.6% (last row). This result indicates the effectiveness of the attention mechanism which combines ECA-Net with CBAM.

## 5. Conclusions

This paper has improved the state-of-the-art object detection method of YOLOv5 for detecting boulders from planetary images. A new feature fusion layer has been added into YOLOv5 to capture more feature information of small boulders. The shallow features have also been brought from the backbone network into feature fusion layers to further reduce the feature information loss of small boulders. Moreover, the attention modules implemented by combining the CBAM and ECA-Net attention mechanisms are also added to highlight the information that contribute to the boulder detection. Based on the VOC2007 dataset which is widely used for detection evaluations and the boulder dataset we have constructed from the images of Bennu asteroid, the evaluation result has shown that YOLOv5 has been improved by 3.4% in precision. In light of the multi-scales of the boulders and the large size of the planetary images, a pyramid based approach is also designed to detect boulders of different scales from different layers of images that have different resolutions. We have applied the proposed approach to detect the boulders from the images of Bennu asteroid. The distribution of the boulders on Bennu asteroid has been also analyzed and presented in this paper.

**Author Contributions:** Conceptualization, L.Z. and C.L.; methodology, L.Z. and C.L.; software, L.Z.; validation, L.Z., X.G. and Z.L.; formal analysis, L.Z.; investigation, L.Z.; resources, X.G.; data curation, L.Z.; writing—original draft preparation, L.Z. and Z.L.; writing—review and editing, C.L. and Z.L.; visualization, X.G.; supervision, Z.L.; project administration, X.G. and C.L.; funding acquisition, C.L. All authors have read and agreed to the published version of the manuscript.

**Funding:** This research was funded by the Open Program of Collaborative Innovation Center of Geo-Information Technology for Smart Central Plains Henan Province, Item number 2020C004.

**Institutional Review Board Statement:** Not applicable.

**Informed Consent Statement:** Informed consent was obtained from all subjects involved in the study.

**Data Availability Statement:** Not applicable.

**Conflicts of Interest:** The authors declare no conflict of interest.

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
