# Peer review of "Improving YOLOv5 with Attention Mechanism for Detecting Boulders from Planetary Images"

_remotesensing, doi:10.3390/rs13183776_

Round 1
Reviewer 1 Report
General aspects
The manuscript describes a specific update and application of an automatized pattern recognition method to identify boulders on planetary surface images using the example of Bennu asteorid. The topic is important for the community and among the main avenue of digital automatized data processing. The structure of the manuscript is moderately good, the language is OK, the illustrations are useful. However there are several mainly minor aspects that should be fixed and explained before publication. As there are several moderate sized corrections the referee suggests moderate or major revision and encourage the authors to improve their manuscript.
- A bit better scientific content is necessary, please put at least one more specific sentence into the introduction mentioning the current main topics with the corresponding citations: boulders are frequent features on solid planetary surfaces (https://ui.adsabs.harvard.edu/abs/2015epl..book.....H/abstract ), especially asteroids, which are being targeted by many missions in the future (https://ui.adsabs.harvard.edu/abs/2018AdSpR..62.1998B/abstract ), where granular and regolith properties are important for landing site selection (https://ui.adsabs.harvard.edu/abs/2014P%26SS..101...65K/abstract ). For example in the case of Bennu asteroid, finding boulder free sampling site was not easy (https://ui.adsabs.harvard.edu/abs/2020Natur.586..484W/abstract , https://ui.adsabs.harvard.edu/abs/2021PSJ.....2..114L/abstract ). Surface feature classification is important not only for landing site selection, but also for various scientific analysis as the number of surveyed structures might be quite large (https://ui.adsabs.harvard.edu/abs/2019JGRE..124..504R/abstract ), where automatized pattern analysis is almost inevitable
- The aim of the usage for bounder identification is useful of course, however on the Moon and Mars (expected frequent targets) rock sizes ranges in a small interval, usually below 50 m diameter, thus not ideal for multi spatial level identification. The situation is better for asteroids, where rock size is larger and maximal spatial resolution is also better. The improved rock identification method might also support better automatized crater identification in the future, which are present with orders of magnitudes larger size range.
- Please clarify in the introduction that the term “detector” used here differs from several other related works, here the detecting method and not the physical instrument was used.
Specific aspects
3 line
“of the geophysical processes”
consider to modify to „of the geological processes”
9 line
„2007( VOC2007)”
put a space after 2007 and delete the space after „(„
25 line
„the geophysical processes”
consider to modify to „the geological processes”
37 line
„images of the planets”
consider to modify to „images of the planetary surfaces”
42 line
„are burred”
do you mean spatial resolution effect or the lack of contrast? please be more specified
46 line
“attention mechanism”
please explain
48 line
“input object”
please explain
56 line
“fusion layer”
please define
93 line
“moon images”
capitalize Moon, as well as in line 95
Figure 1.
Are the acronyms like CSP explained somewhere? at least in a cited reference?
154 line
“152*152, 76*76, 38*38, and 19*19”
are these values pixel based sizes? if yes, indicate
167 line
„76*76*255, 38*38*255, and 19*19*255”
are the last number the binary coding? if yes, indicate
172 line
„pooling”
explain briefly for unexperienced readers
187 line
„width of the feature map”
in pixels?
207 line
„but effectively captures information about cross-channel interactions”
please briefly explain
223 line
„(2)the”
226 line
„ (3)and”
put spaces after the numbers
Fig 2.
„there are three improvements in the architecture”
is it marked with the green background? if yes, please indicate
265 line
„Therefore, we have removed these slices in which there are none boulders.”
Did this happen manually? Were such non-boulder images used for control measurements?
278 line
„Adam optimizer”
if there is a corresponding reference, please cite here
283 line
why is FPS important? please explain
286-287 lines
please clarify this sentence, not easy to understand
Table 2
please indicate briefly in the caption, what are the differences between the listed detectors (first column)
317-318
„then detect the objects from each layer”
using different spatial resolution?
329 line
„detector(both”
put a space
334
„The third step is the boulder detection process in which the boulders are detected by the improved YOLOV5 detector.”
based on what? pattern? what type of pattern?
380
were there overlap between the nearby individual slices?
Table 3
please consider to give an extra column with the physical size
355
„of Bennu asteroid”
with how much km2 area?
Figure 5 caption
„of Bennu asteroid”
do you mean slice size?
361
„the large boulders”
please give diameter
„3000 square metres”
give diameter also
5.2 subtitle
please use „Boulder distribution statistics”
Figure 6
give absolute size too at the horizontal axis
358
„34,472 boulders detected in the 40°N-88°N region, 71,299 boulders in the 0-40°N region, 87,065 boulders in the 0-45°S region, and 64,402 boulders in the 45°S-91°S region.”
please give absolute sizes of the areas and also boulder/km2 values
Figure 8
please give the projection type
Reviewer 2 Report
Please see comments in PDF.

Author Response
Please see the attachment.

This manuscript is a resubmission of an earlier submission. The following is a list of the peer review reports and author responses from that submission.